# Tracking the Origins of *Pseudomonas aeruginosa* Phylogroups by Diversity and Evolutionary Analysis of Important Pathogenic Marker Genes [†]

Sara E. Quiroz-Morales [1], Selene García-Reyes [1], Gabriel Yaxal Ponce-Soto [2], Luis Servín-González [1] and Gloria Soberón-Chávez [1,*]

[1] Departamento de Biología Molecular y Biotecnología, Instituto de Investigaciones Biomédicas, Universidad Nacional Autónoma de México, Tercer Circuito Exterior, Ciudad Universitaria, Apdo, Postal 70228, Mexico City 04510, Mexico; squiroz@iibiomedicas.unam.mx (S.E.Q.-M.); selenegarcia@iibiomedicas.unam.mx (S.G.-R.); servin@iibiomedicas.unam.mx (L.S.-G.)

[2] Microbial Paleogenomics Unit, Department of Genomes & Genetics, Institut Pasteur, 75015 Paris, France; gabriel.ponce-soto@pasteur.fr

[*] Correspondence: gloria@iibiomedicas.unam.mx

[†] This article is dedicated to the memory of Ulises de Jesús Ortiz, deceased on 5 November 2021.

**Abstract:** *Pseudomonas aeruginosa* is a widespread environmental bacterium and an opportunistic pathogen that represents a health hazard due to its production of virulence factors and its high antibiotic resistance. The genome of most of the strains belonging to this bacterial species is highly conserved, and genes coding for virulence-associated traits are part of the species core-genome. Recently, the existence of phylogroups has been documented based on the analysis of whole genome sequences of hundreds of isolates. These clades contain both clinical and environmental strains, which show no particular geographical distribution. The major phylogroups (clades 1 and 2) are characterized by the nearly mutually exclusive production of the virulence effectors secreted by the type three secretion system (T3SS) ExoS and ExoU, respectively. Clade 3 is the most genetically diverse and shares with clade 5, which is closely related to clades 1 and 2, the production of the pore-forming exolysin A, and the lack of T3SS, among other characteristics. Here we analyze the 4955 *P. aeruginosa* genomes deposited in the Pseudomonas Genome Database and present some hypotheses on the origins of four of the five phylogroups of this bacterial species.

**Keywords:** *Pseudomonas aeruginosa* phylogroups; type 3 secretion system; *exoS*; *exoU*; exolysin A; pyocyanin; Pseudomonas Genome Database





## 1. Introduction

*Pseudomonas aeruginosa* is a widespread environmental bacterium [1] that has even been claimed to be ubiquitous [2] and represents a critical health hazard due to its production of virulence factors [3–5] and high intrinsic and acquired antibiotic resistance [6,7].

The genomic constitution of this bacterial species is characterized by the high degree of genome conservation among clinical and environmental isolates that have different geographical origins and times of isolation, as well as the conservation among all isolates– both environmental and clinical–of the virulence-associated traits [8,9]; thus, all strains are potentially pathogenic. These highly conserved virulence factors comprise proteases such as elastase LasB, toxins such as pyocyanin (PYO), and the biosurfactant rhamnolipids (RL), among others, that are coordinately expressed by the complex regulatory circuit called quorum sensing (QS) [10] and the type III secretion system (T3SS), which translocates effector proteins into the host cells [11]. Furthermore, it has been reported by the genome analysis of *P. aeruginosa* isolates from different habitats that the genetic repertoire of clones maintains a habitat-independent gradient of virulence [12].

In 2010, the existence of a group of *P. aeruginosa* strains represented by the clinical isolate PA7, that are genetically diverse, lack the T3SS, and produce a cytotoxic pore-forming exolysin (ExlA) was reported [13]. In 2015, the analysis of the whole genome sequence of 389 strains [14] revealed that this bacterial species was constituted by three phylogroups: Clade 1 contained most of the studied strains (309), including the PAO1 type strain, as well as the C clone which is the most frequently isolated *P. aeruginosa* lineage [15]; clade 2, which is closely related to the first group, and was comprised by 68 strains including its type strain PA14; 12 strains were found to belong to the genetically diverse clade 3 with PA7 as the type strain.

The most apparent distinct characteristic of strains belonging to clades 1 and 2 is that most of the strains that belong to the first phylogroup produce the T3SS effector ExoS, while most of those belonging to clade 2 produce the T3SS effector ExoU [16,17]. Only around 1% of the strains belonging to either of these groups, although mostly belonging to clade 1, produce both ExoS and ExoU effectors [16].

In 2019, two articles addressed the phylogenetic characterization of *P. aeruginosa* populations, including clinical and environmental isolates [16,17]. In both cases, the authors detected a fourth phylogenetic group that lacks the T3SS and produces the ExlA exolysin. This phylogroup, called group 5 by Freschi et al. [17], is more closely related to clades 1 and 2 than clade 3. Clade 5 shares different characteristics with clade 3 besides its lack of T3SS and the production of ExlA, such as the inability to produce RL with two rhamnose moieties (di-RL)–because strains of both clades lack *rhlC*, which encodes the rhamnosyl transferase that converts mono-RL to di-RL [18]–and the inability to produce the phenazine-1-carboxamide (PCN)–because they lack the gene encoding the PhzH enzyme that produce it, using phenazine-1-carboxylic acid (PCA) as substrate, which is also the substrate for the synthesis of PYO by a diverging route [19].

The aim of this work is the bioinformatic analysis of the chromosomal regions encoding the distinct characteristics of each of the phylogenetic groups reported by Olzer et al. in 2019 [16] and Freschi et al. in 2019 [17], using representative strains to define marker sequences, that were then used to classify the 4955 *P. aeruginosa* genomes deposited in the Pseudomonas Genome Database (PsGenDatab, www.pseudomonas.com; accessed on 15 May 2021), which includes both clinical and environmental isolates [20]. We used two valuable resources as a starting point for our analysis: on one hand, the huge amount of information organized and curated in PsGenDatab, and on the other, the whole genome analysis of hundreds of strains already reported [16,17] that defined the *P. aeruginosa* pangenome [17], and which allowed us to track specific genetic markers related to pathogenicity that were defined in these works. The results obtained by this focused analysis enabled us to propose hypotheses on the origins of four of the five *P. aeruginosa* phylogroups. In addition, we tried to follow a similar strategy to detect strains belonging to the group defined by Freschi et al. in 2019 [17] as group 4, which was identified by whole sequence analysis, however the focused strategy presented here did not result in the identification of genetic markers that could be used in tracking clade 4 among the genomes deposited in PsGenDatab.

## 2. Materials and Methods

### 2.1. Analysis of Genome Sequences

The PsGenDatab (www.pseudomonas.com; accessed on 15 May 2021) was used to obtain genome sequences used in this work. The sequences for the genes *exoS* and *exoT* of PAO1, as well as the PA14_14300 gene from PA14, were retrieved from PsGenDatab. A set of primers were designed with SnapGene viewer version 6.0.2 (GSL Biotech LLC, San Diego, CA, USA; available at snapgene.com) to amplify the *exoS* gene. Given the highly similarity between *exoS* and *exoT*, global nucleotide alignments were carried out with Clustal Omega [21] at the EMBL-EBI web server to identify the gene regions shared between both genes and exclude them during the primer design process. The search of *exoS* among the genomes deposited in the PsGenDatab (Table S1) was done by virtual PCR with

SerialCloner version 2.6.1 (Serial Basics; available at http://perezf.free.fr/SerialBasics/Serial_Cloner.html). The oligonucleotides used for these virtual PCR reactions are: Rv-exoS GTATCCAAGGCGAGCAACCTG and Fw-exoS CGGGAGATCGAGAGCGAGAAAAA with an expected amplification size of 1893 bases. The expected amplification sequence was subsequently used for a Blast search in the PsGenDatab using the default search parameters and including only complete genomes. A total of 3487 non-redundant strains belonging to the clade 1 were identified, considering a sequence length of 1893 bp and an identity equal or higher than 98%.

To identify the strains belonging to clade 2, the absence of the *exoS* gene together with the presence of gene PA14_14300 in the vicinity of *spcS* was considered. As previously described, a Blast search was performed in the PsGenDatab with default parameters. A total of 1394 non-redundant strains with an identity equal or higher than 97% were identified. *P. aeruginosa* strains PAO1, PA14, PA7 y PA-W1 were used as references for clades 1, 2, 3, and 5, respectively.

### 2.2. Determination of Average Nucleotide Identity (ANI) Index among P. aeruginosa Genomes

The computation of the Average Nucleotide Identities (ANI) was performed for some strains belonging to clades 1, 2, 3, and 5. For clade 1, the selected strains were PAO1 (GCF_000006765.1), LESB65 (GCF_000583955.1), WH-SGI-V-07678 (GCF_001453545.1), and PAK (GCA_000568855.2). The representatives for clade 2 were PA14 (GCF_000014625.1), AZPAE15016 (GCF_000792965.1), ID4365 (GCA_000647615.1), VET-35 (GCF_003630095.1), and PA_W32 (GCA_003833565.1). In the case of clade 3, the included strains were PA7 (GCA_000017205.1), ATCC9027 (GCA_001374975.1), LMG5031 (GCA_003837245.1), and CLJ1 (GCA_003032395.1). For clade 5, we analyzed PA-W1 (GCA_003833685.1), CF_PA39 (GCA_000568235.2), ENV-567 (GCA_003633495.1), and 014-2A (GCA_003835685.1). The ANI values were calculated using fastANI v1.33 [22].

### 2.3. Construction of Phylogenetic Trees

To construct the phylogenetic tree, we used the genome sequences of *P. aeruginosa* members from clades 1 (shown in blue) and 2 (shown in purple), and candidates of strains belonging to clade 4 (shown in green), as well as PA7 used as outgroup that were retrieved from the NCBI. The analyzed strains correspond to 197S020911BSL_PA3 (GCA_004375175.1), AUS4085 (GCA_003840655.1), B136-33 (GCA_000359505.1), Cotonu1 (GCF_003836245.1), CPHL6749 (GCF_003836645.1), CPHL8203 (GCF_003837525.1), CPHL9433 (GCF_003698765.1), F69A_isolate_IST27N (GCA_003698655.1), ID4365 (GCA_000647615.1), LESB58 (GCF_000026645.1), LMG2107 (GCF_003836355.1), M10 (GCF_000647655.1), Mi159 (GCF_003836315.1), Mi162_2 (GCF_003698835.1), PA_D25 (GCA_001721845.1), PA102 (GCA_003332585.1), PA14 (GCF_000014625.1), PA7 (GCF_000017205.1), PAC13B (GCF_004373995.1), PAC15B (GCA_004373515.1), PAC9A (GCF_004373985.1), PAK (GCA_000408865.1), PAO1 (GCF_000006765.1), PA-W24 (GCF_003834125.1), PA-W32 (GCF_003833565.1), PA-W34 (GCF_003841525.1), PA-W36 (GCA_003833545.1), PA-W4 (GCF_003833785.1), PA-W45 (GCF_003833425.1), PA-W8 (GCF_003841755.1), PMM38 (GCF_003836135.1), RP1 (GCA_001500235.1), strain 19660 (GCA_000481765.1), Tu61 (GCA_003836685.1), TuD199 (GCF_003836175.1), W5Aug16 (GCF_003836445.1), WH-SGI-V-07166 (GCA_001449485.1), and WH-SGI-V-07252 (GCA_001452885.1). The genomes were aligned to identify locally collinear blocks (LCB) using Mauve [23] with the progressive Mauve algorithm [24]. Subsequently, the LCBs shared by all genomes with a minimum length of 500 bases were extracted and concatenated, resulting in an alignment of 6,103,417characters. Phylogenetic reconstruction was performed with FastTree2 [25] using the GTR model of nucleotide substitution and 1000 bootstraps.

## 3. Results and Discussion

### 3.1. Analysis of the Genomes of Strains Belonging to Clades 1 and 2

To obtain information regarding the chromosomal differences of strains belonging to clades 1 and 2, we analyzed the chromosomal regions that surround the *exoS* (PA3841) and *exoU* (PA14_51530) genes of the type strains PAO1 and PA14, respectively, since these are the most conspicuous markers of these two phylogroups, and since it has been proposed that these lineages are genetically isolated [16].

We found that PA14 shows a deletion of the *exoS* gene present in the PAO1 chromosome (corresponding to PA3841) but retains *spcS* (PA14_14330), which encodes the chaperone required to secrete ExoS. In addition, PA14 contains three genes adjacent to *spcS* that are not present in the PAO1 chromosome: PA14_14320 (encoding a hypothetical protein), PA14_14310, and PA14_14300. The last two genes seem to constitute an operon (Figure 1) and encode a transcriptional regulator and a zinc-binding oxidoreductase, respectively (www.pseudomonas.com; accessed on 15 May 2021). We did not detect any sign of a transposase or of an insertion sequence that might represent a vestige of the chromosomal rearrangement involved in the deletion of *exoS* in the PA14 chromosome.

The PA14 *exoU* and *spcU* genes form part of the 10.7 kb pathogenicity island PAPI-2 [26], and in the corresponding position, strain PAO1 contains a 8.9 kb genomic island that harbors, among others, the gene encoding pyocin S5 (*pyoS5*) [26] (Figure 1). The presence of T3SS effectors in mobile genetic elements might be a common phenomenon since it has been reported that in the plant pathogenic bacterium *Pseudomonas syringae*, an integrative conjugative element (ICE) containing loci encoding multiple type III effectors can conjugate between different strains and modify their interaction with plants [27].

It is interesting that the clade 2 specific ExoU production is encoded in a mobile genetic element since it could be expected that this would lead to genomic instability that could hardly result in the perpetuation of the second-largest *P. aeruginosa* phylogroup. A plausible explanation for the stable maintenance of ExoU production in clade 2 strains is that the production of this exotoxin encoded in the PAPI-2 pathogenicity island is an important virulence trait [28] that is under positive selection. In this respect, it has been reported that ExoU is the most toxic of the T3SS effectors [29]. Another requisite for the fixation of clade 2 genomic constitution is that recombination between this phylogroup and clade 1 is low, as has been reported previously [16].

Strain PA14 contains another pathogenicity island, PAPI-1, that is larger than PAPI-2 and is not present in strain PAO1 [26,28]. However, this pathogenicity island is a mobile genetic element [30] that is present in strains belonging to both clades 1 and 2, and thus is not phylogenetically relevant.

We decided not to use *exoU* and other genes encoded in the PAPI-2 pathogenicity island to screen for strains belonging to clade 2 among the 4955 *P. aeruginosa* genomes in the PsGenDatab to avoid the use of a gene encoded in a transposable element. Instead, we decided to use PA14_14300, one of the genes near *spcS* in the PA14 chromosome that is absent in PAO1 (Figure 1). To validate whether PA14_14300 was a good marker for strains belonging to clade 2, we determined the presence of this gene among 183 strains that contain *exoU* and lack *exoS*–which were reported to belong to this phylogroup–using a whole genome phylogenomic approach [16], and found the gene in all of them.

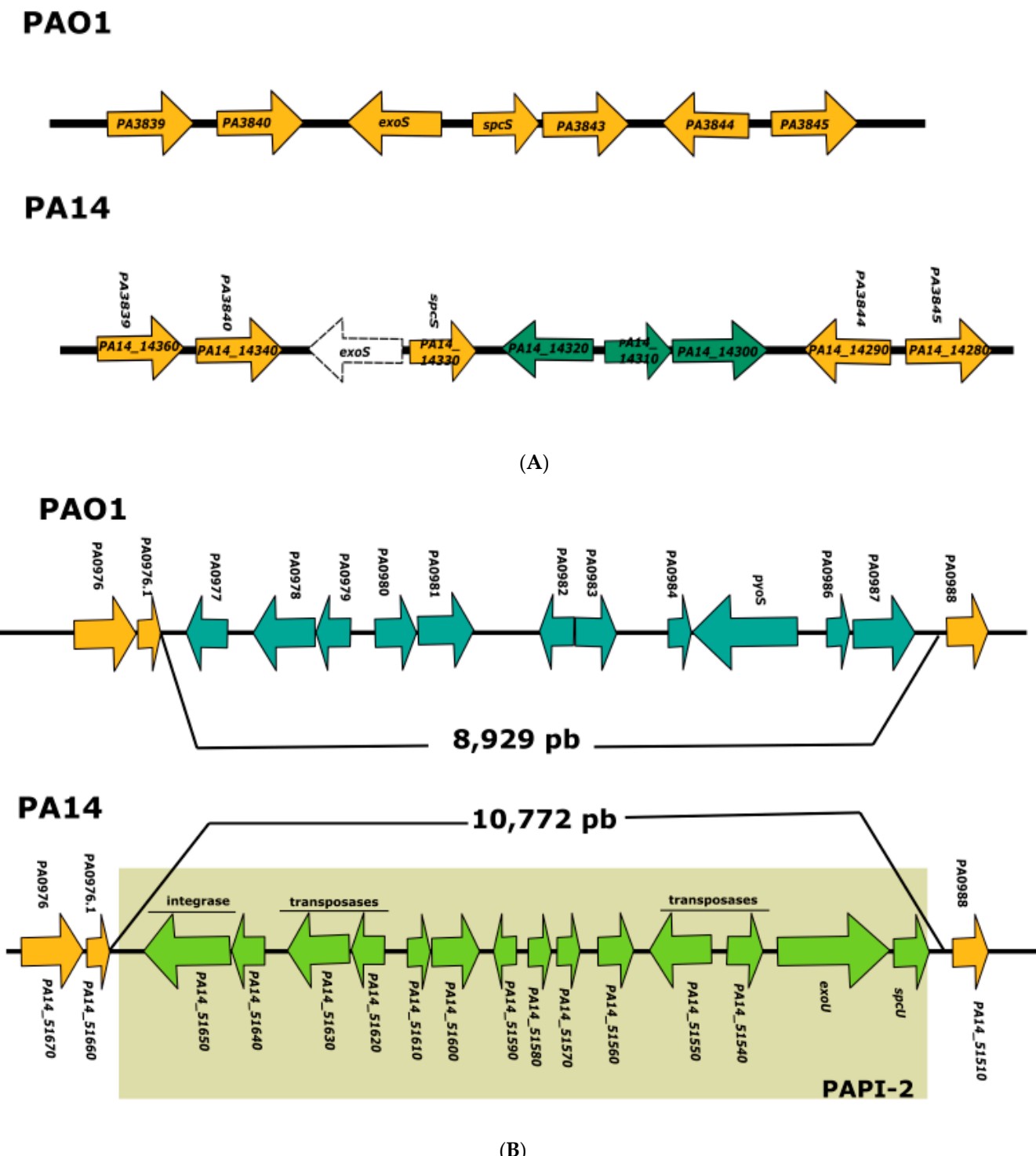

(**A**)

(**B**)

**Figure 1.** **Analysis of PAO1 and PA14 genome regions where *exoS* and *exoU* are encoded.**
(**A**) Representation of the PAO1 and PA14 chromosomal regions where *exoS* is encoded in the
former strain; homologous genes are shown in orange; PA14_14300, PA14_14310, and PA14_14320
genes that are present only PA14 chromosome are shown in green, and the deletion of *exoS* in PA14
is shown by a dotted line. (**B**) Representation of the PAO1 and PA14 chromosomal regions where
*exoU* is encoded in the latter strain; homologous genes are shown in orange, genes corresponding
to PAPI-2 pathogenicity island are shown in green, and genes corresponding to the 8.9 kb PAO1
genomic island are shown in blue.

To track the strains belonging to clades 1 and 2 in the 4955 *P. aeruginosa* genomes in the PsGenDatab, we carried out virtual PCR for exoS and a BLAST search for PA14_14300 nucleotide sequences and found that 3487 genomes (70.4%) contained *exoS* and thus presumably belong to clade 1 (see Supplementary Material Table S1), while 1394 genomes (28.1%) lacked *exoS* and contained PA14_14300, and thus seem to belong to clade 2 (see Supplementary Material Table S2). This proportion of strains belonging to each of these two clades is similar to that reported in the articles analyzing the genome sequences of hundreds of *P. aeruginosa* strains [16,17].

The number of genomes in the database that we detected using this approach as belonging to clades 1 and 2 is lower than the total number of deposited genomes, (4883 vs. 4955). This is due in part to the small number of genomes that belong to clades 3 and 5 that lack both *exoS* and PA14_14300 (see below), and probably also to some inconsistencies of the PsGenDatab which contains draft genomes and some duplicated entries. In addition, it is important to consider that there is a small proportion of strains that might be rare recombinants of the two clades.

To determine the presence of recombinants between clades 1 and 2, we searched for strains that contained both *exoS* and *exoU* and found 49 strains (Table 1)–around 1% of the analyzed strains–which is similar to the proportion that has been reported previously [16]. Among these 49 strains, eight contain the PA14_14300 gene (although three of them contain only a fragment)–suggesting that they belong to clade 2 and inherited the *exoS* gene by recombination with a strain from clade 1–while the 41 strains that lack PA14_14300 gene presumably belong to clade 1. However, since the strains containing both *exoS* and *exoU* are clearly the product of recombination between members of clades 1 and 2, the use of a single genetic marker to assign a phylogroup is questionable, and a global approach should be performed to classify them. To exemplify this phenomenon, we determined the ANI value of the eight strains that contain the PA14_14300 gene among the strains that contain both *exoS* and *exoU and* found that four of them are more closely related to strains of clade 1 than of clade 2 (see Supplementary Material Table S3). That most of the strains containing *exoS* and *exoU* belong to clade 1 is consistent with the analysis reported previously [16].

**Table 1.** Strains encoding both ExoS and ExoU in their genome that are deposited in the PsGenDatab.

| STRAIN | PA14_51550 (PAPI-2) | PA14-14300 | Source of Isolation | Place of Isolation | Accession Number |
|---|---|---|---|---|---|
| WH-SGI-V-07678 | + | − | Hospital | United States | GCF_001453545.1 |
| AZPAE14698 | + | − | Respiratory tract infection | Israel | GCF_000794705.1 |
| AZPAE15028 | + | − | Respiratory tract infection | France | GCF_000792265.1 |
| AZPAE15028 | + | − | Unknown | United States | GCF_007021965.1 |
| PA99 | + | − | Urine | United States | GCF_000611995.2 |
| AUS122 | + | − | River, environmental | Australia | GCF_003976545.1 |
| T4242 | + | − | Sputum | Thailand | GCF_003975125.1 |
| MRSN1739 | + | − | Blood | United States | GCF_003969695.1 |

**Table 1.** *Cont.*

| STRAIN | PA14_51550 (PAPI-2) | PA14-14300 | Source of Isolation | Place of Isolation | Accession Number |
|---|---|---|---|---|---|
| ENV-208 | + | + | Residual water, environmental | Estonia | GCF_003633235.1 |
| VET-77 | + | − | Dog ear secretion | Estonia | GCF_003631375.1 |
| VET-39-D2 | + | − | Dog ear secretion | Estonia | GCF_003631365.1 |
| VET-44 | + | − | Ear secretion | Estonia | GCF_003631355.1 |
| PABL031 | + | − | Blood | United States | GCF_003411975.1 |
| L25 | + | − | Hospital effluent | Brazil | GCF_003402335.1 |
| M12 | + | − | Hospital effluent | Brazil | GCF_003402275.1 |
| WCHPA075056 | + | − | Clinical isolate | China | GCF_002976215.1 |
| WCHPA075063 | + | − | Clinical isolate | China | GCF_002976195.1 |
| CCF_716 | + | − | Clinical isolate | N/A | GCF_001910215.1 |
| WH-SGI-V-07638 | + | − | Hospital | United States | GCF_001453435.1 |
| AZPAE12423 | + | − | Cystic Fibrosis | Cleveland, United States | GCF_000797355.1 |
| AZPAE12418 | + | − | Cystic Fibrosis | Cleveland, United States | GCF_000797245.1 |
| AZPAE12416 | + | − | Cystic Fibrosis | Cleveland, United States | GCF_000797185.1 |
| AZPAE14899 | + | − | Urinary tract infection | Chennai, India | GCF_000791065.1 |
| ATCC 25324 | Incomplete genome | + | Glass shredder, air, environmental | Unknown | GCF_000297295.1 |
| 6093 | + | − | Cystic Fibrosis, sputum | Quebec, Canada | GCF_004371325.1 |
| MRSN16344 | + | − | Wound | United States | GCF_003969715.1 |
| AUS476 | + | − | Cystic Fibrosis | Australia | GCF_003840865.1 |
| CN573=PSE143 | + | − | Pleural liquid | Georgia | GCF_003836905.1 |
| A1(A2448) | + | − | Urine | Thailand | GCF_003835445.1 |
| HUM-257 | + | − | Bronchus-alveolar wash | Estonia | GCF_003631435.1 |
| M28A1 | + | − | Cow manure | Colombia | GCF_002117025.1 |
| BK5 | + | − | Eye, keratitis | Madurai, India | GCF_002242915.1 |
| isolate 406 | + | + | Unknown | EMBL | GCF_900147325.1 |
| isolate 405 | + | + | Unknown | EMBL | GCF_900147315.1 |
| 1046_PAER | + | − | Clinical isolate | United States | GCF_001060185.1 |
| AZPAE14960 | + | + | Respiratory tract infection | Spain | GCF_000791765.1 |
| 1163 | + | − | Blood | Brazil | GCF_009857765.1 |
| PA126 | + | − | Eye, keratitis | Australia | GCF_009727515.1 |
| AZPAE15052 | + | − | Respiratory tract | Argentina | GCF_000791465.1 |
| AZPAE14930 | Fragment | + | Urinary tract infection | Germany | GCF_000793865.1 |

**Table 1.** *Cont.*

| STRAIN | PA14_51550 (PAPI-2) | PA14-14300 | Source of Isolation | Place of Isolation | Accession Number |
|--------|---------------------|------------|---------------------|--------------------|------------------|
| HM293 | Fragment | − | Bladder cancer | United Kingdom | GCF_003835395.1 |
| VET-35 | Fragment | + | Dog ear secretion | Estonia | GCF_003630095.1 |
| S700_C14_RS | Fragment | − | Respiratory tract infection | Italy | GCF_002135765.1 |
| PABL081 | Fragment | − | Blood | United States | GCF_003411045.1 |
| PABL075 | Fragment | + | Blood | United States | GCF_003410855.1 |
| PABL086 | Fragment | − | Blood | United States | GCF_003410925.1 |
| S708_C14_RS | Fragment | − | Respiratory tract infection | Italy | GCF_002136415.1 |
| PABL074 | Fragment | − | Blood | United States | GCF_003411025.1 |
| PA176 | Fragment | − | Eye | Australia | GCF_009727405.1 |

The presence of the PAPI-2 pathogenicity island among the 49 *exoU*, *exoS* recombinant strains was determined by a BLAST search using as query the sequence of PA14_51550, the gene encoding a transposase that flanks *exoU*. We found that all these 49 strains contain the PAPI-2 pathogenicity island, but three strains–the blood isolate 1163 from Brazil, PA126 strain isolated from an eye infection patient in Australia, and strain AZPAE 15052 isolated from the respiratory tract of a patient in Argentina (Table 1)–contain additional genes encoding transposases within this pathogenicity island (www.pseudomonas.com; accessed on 15 May 2021).

This analysis of the genomic regions surrounding the *exoS* and *exoU* genes among the PAO1 and PA14 type strains, and the correlation of these arrangements among the genomes of around 5000 strains that have been deposited in the PsGenDatab, allowed us to formulate the following hypotheses for the origin of the two major *P. aeruginosa* phylogroups: the clade 2 ancestor strain (belonging to clade 1) suffered a deletion of *exoS* and the concomitant acquisition of three genes that do not seem be involved in pathogenicity or fitness (PA14_14320, PA14_14310, and PA14_14300). In addition, this strain acquired by horizontal gene transfer (HGT) a 10 kb pathogenicity island encoding *exoU* and *spcU*, which is present in a different chromosomal region [26].

A central part of this hypothesis is that when clade 2 diverged from clade 1, the resulting lineages presented a very low frequency of recombination so the clone giving rise to this new phylogroup could become fixed. This low frequency of recombination between clades 1 and 2 is the basis for the delimitation of these lineages, as has already been reported [16] and agrees with the results reported here analyzing PsGenDatab genome sequences. However, it is important to stress the low genetic variability of these *P. aeruginosa* phylogroups [8], which is evidenced by the high conservation of their genomic sequences not only in coding sequences but also in intergenic regions; clade 1 presents an ANI value of more than 99% between strains, and of more than 98% with clade 2, Table 2 [16]. This DNA sequence conservation seems to be the product of a gene conversion mechanism, which operates on both clades 1 and 2, although the frequency of interclade recombination is lower than that of intraclade genetic exchange, giving rise to two closely related phylogroups. The existence of the mechanism of gene conversion can be appreciated considering that the four strains belonging to clade 1, presented as examples in Table 2, have an ANI value of more than 99%, and a different geographical origin (Australia, United States, and the United Kingdom) and time of isolation (that can be of decades apart).

**Table 2.** Average nucleotide identity (ANI) of some genomes belonging to phylogroups 1 (blue), 2 (pink), 3 (red), and 5 (green) *.

| STRAINS | 1 | 2 | 3 | 4 | 5 | 6 | 7 | 8 | 9 | 10 | 11 | 12 | 13 | 14 | 15 | 16 |
|---|---|---|---|---|---|---|---|---|---|---|---|---|---|---|---|---|
| (1) PAO1 | 100 | | | | | | | | | | | | | | | |
| (2) LESB65 | 99.35 | 100 | | | | | | | | | | | | | | |
| (3) PAK | 99.20 | 99.15 | 100 | | | | | | | | | | | | | |
| (4) WH-SGI-V-07678 | 99.29 | 99.15 | 99.07 | 100 | | | | | | | | | | | | |
| (5) PA14 | 98.73 | 98.63 | 98.62 | 98.66 | 100 | | | | | | | | | | | |
| (6) ID4365 | 98.71 | 98.62 | 98.54 | 98.59 | 98.96 | 100 | | | | | | | | | | |
| (7) PA_W32 | 98.68 | 98.50 | 98.59 | 98.55 | 98.95 | 98.83 | 100 | | | | | | | | | |
| (8) AZPA E15016 | 98.69 | 98.56 | 98.58 | 98.57 | 99.95 | 98.93 | 98.85 | 100 | | | | | | | | |
| (9) PA7 | 93.93 | 93.91 | 93.86 | 93.88 | 93.90 | 94.01 | 93.84 | 93.88 | 100 | | | | | | | |
| (10) ATCC 9027 | 93.80 | 93.73 | 93.67 | 93.69 | 93.75 | 93.85 | 93.81 | 93.74 | 98.95 | 100 | | | | | | |
| (11) CLJ1 | 93.81 | 93.76 | 93.64 | 93.71 | 93.72 | 93.65 | 93.83 | 93.71 | 98.74 | 99.21 | 100 | | | | | |
| (12) LMG 5031 | 93.80 | 93.73 | 93.72 | 93.71 | 93.75 | 93.80 | 93.84 | 93.72 | 98.88 | 99.24 | 99.11 | 100 | | | | |
| (13) PA-W1 | 97.46 | 97.44 | 97.35 | 97.40 | 97.28 | 97.35 | 97.34 | 97.24 | 93.80 | 93.70 | 93.74 | 93.71 | 100 | | | |
| (14) CF_PA39 | 97.50 | 97.47 | 97.36 | 97.44 | 97.29 | 97.34 | 97.30 | 97.26 | 93.84 | 93.75 | 93.77 | 93.71 | 99.50 | 100 | | |
| (15) 014-2A | 97.44 | 97.43 | 97.35 | 97.34 | 97.27 | 97.30 | 97.27 | 97.21 | 93.79 | 93.72 | 93.67 | 93.74 | 99.96 | 99.49 | 100 | |
| (16) ENV-567 | 97.45 | 97.38 | 97.34 | 97.37 | 97.26 | 97.31 | 97.28 | 97.16 | 93.80 | 93.67 | 93.69 | 93.69 | 99.49 | 99.49 | 99.44 | 100 |

* The strains analyzed were numbered, and the corresponding number is used at the top of the table. Genome sequences were obtained from PsGenDatab.

The molecular mechanism involved in the lower frequency of interclonal recombination is not clear and could be multifactorial but does not seem to be related to the role of ExoS and ExoU in pathogenicity or fitness, since strains exhibiting both markers are readily isolated (Table 1). One factor that could reduce recombination between clade 1 and 2 would be that the genetic exchange between strains belonging to each phylogroup in biofilms is reduced. A large part of the *P. aeruginosa* biofilm matrix is extracellular DNA (e-DNA) produced by explosive cell lysis [31] caused by pyocins R and F [32], and it has been reported that this bacterial species is capable of natural transformation in biofilms [33], and that HGT occurs in these structures [34]. Thus, if the frequency of mixed biofilms containing strains of both clades is low, the frequency of recombination will be reduced. This possibility remains to be experimentally validated.

Another possibility to explain the lower frequency of interclonal recombination is that very efficient restriction systems exist in these phylogroups in such a way that strains that belong to one clade efficiently restrict the DNA of the other clade, and vice versa.

### 3.2. Analysis of the Genomes of Strains Belonging to Clade 3

Strains belonging to clade 3 are genetically diverse (Table 2) and thus have been considered an outlier group [13,16,17]. The most conspicuous characteristic of all strains belonging to this diverse clade is that they present the pore-forming exolysin A (ExlA)– which is an important virulence factor of this phylogroup–and the concomitant deletion of genes encoding the T3SS [35] and the effectors secreted by this system like ExoS and ExoU. It is important to highlight that the production of ExlA and the presence of the T3SS are mutually exclusive and that no *P. aeruginosa* isolate has been found to have genes encoding for both virulence-associated traits. Therefore, one possibility to explain the origin of clade 3 would be that a *P. aeruginosa* strain inherited by HGT–the *exlBA* operon that encodes

for ExlB the outer membrane translocator of ExlA and the exolysin A itself [36]–and that to survive or to maintain its virulence, the deletion of the T3SS was selected. The *exlBA* operon is present in several *Pseudomonas* species, like *Pseudomonas fluorescens*, *Pseudomonas putida* and *Pseudomonas chlororaphis* [36], and one of these could be the source of the operon encoding exolysin A. In addition, it has been reported that members of clade 3 contain a deletion of *rhlC*–encoding the enzyme involved in the production of di-RL [18]–and of *phzH*–encoding the enzyme that transforms PCA, the PYO precursor, to PCN [19].

The analysis of the PA7 genome (Figure 2) showed that the 36 genes encoding proteins involved in the T3SS were deleted, as was reported previously [13,35]. This deletion generated a 56 bp sequence, annotated in the PA7 genome as the intergenic region between PA1689 and *bglX* (Figure 2), which shows no homology to the PAO1 chromosomal region flanking the 36 genes involved in T3SS synthesis. We used this 56 bp sequence to search the PsGenDatab and found 36 strains among the 4955 *P. aeruginosa* genomes (see Supplementary Material Table S4) that contain it (Figure 2). All of them contained the *exlBA* operon in the same chromosomal position as strain PA7.

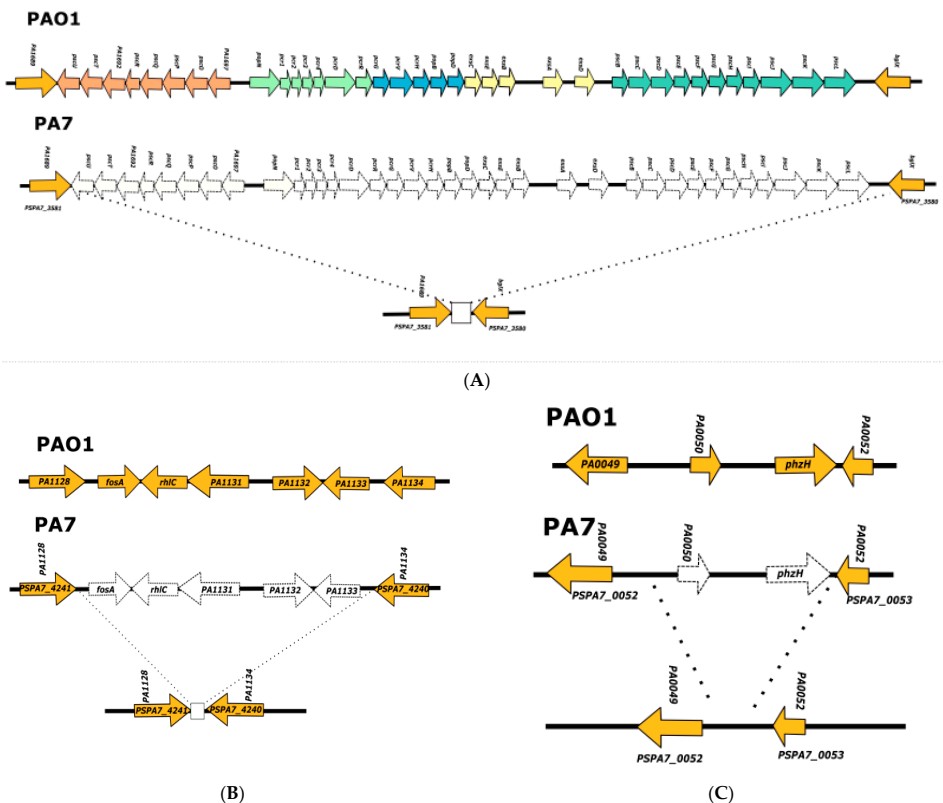

**Figure 2. Analysis of PA7 genome regions where the genes involved in T3SS, *rhlC,* and *phzH* are encoded in the PAO1 strain.** (**A**) Representation of the PAO1 and PA7 chromosomal regions, where genes involved in the T3SS are encoded in the former strain; the five operons encoding genes involved in the T3SS in the PAO1 chromosome are shown in different colors, and the lack of the 36 genes encoded in these operons in strain PA7 chromosome is shown with a dotted line; the square at the bottom of the figure represents the 56 bp sequence that is present in the 36 strains belonging to clade 3 detected in the PsGenDatab. (**B**) Representation of the PAO1 and PA7 chromosomal regions where *rhlC* is encoded in the former strain; the homologous genes are shown in orange, and the genes that are absent in the PA7 chromosome are shown with a dotted line; the square shown at the bottom of the figure corresponds to the 93 bp sequence that is present in the 36 strains belonging to clade 3 detected in the PsGenDatab. (**C**) Representation of the PAO1 and PA7 chromosomal regions where *phzH* is encoded in the former strain; homologous genes are shown in orange and genes that are absent in the PA7 chromosome are shown with dotted lines.

The analysis of the PA7 genome sequence corresponding to the *rhlC* locus showed that this strain contains a deletion of the entire gene–the *fosA* gene flanking *rhlC* on one side, and three genes (corresponding to PA1131, PA1132, and PA1133 in PAO1) flanking it on the other side (Figure 2). We determined that a 93 bp sequence with no homology to PAO1 was present in the PA7 chromosome in the region where *rhlC* and its flanking genes were deleted. Using this 93 bp sequence (annotated as the intergenic region between PA1128 and PA1134, Figure 2), we detected the same 36 *exlBA* positive strains that were detected using the 56 bp sequence, that is, the scar of the deletion of genes encoding proteins involved in T3SS (Table S4).

A similar approach was performed with the sequence that is present in the PA7 chromosome delimiting the boundaries of the deletion of *phzH* and one flanking gene (corresponding to the PA0050 PAO1 gene) (Figure 2), and the same 36 strains were detected (Table S4). These 36 include several strains that have been reported previously as belonging to the PA7 phylogroup [35].

These results suggest that the genetically diverse clade 3 originated when the *exlBA* operon was inherited by HGT by a *P. aeruginosa* strain [36], and that deletion of genes encoding for the T3SS, RhlC, and PhzH was concomitantly selected.

The high degree of genetic divergence of clade 3 strains compared with the conservation of the other clades (an ANI value slightly higher than 93%, Table 2) [16,17] suggests that they have been in recombination isolation from the most abundant clades 1 and 2 for a long period; presumably, since they first originated.

It has been reported that the global transcriptional regulator of *P. aeruginosa* PAO1 and PA14 genes involved in virulence–Vfr (an ortholog of the *Escherichia coli* catabolite repressor protein-CRP) coupled to cyclic AMP (cAMP)–regulates the expression of the *exlBA* operon [37,38]. In addition, it has been shown that the QS-dependent expression of different virulence traits like RL, PYO, and elastase LasB is conserved in the ATCC 9027 strain belonging to clade 3 [39,40]. The conservation in clade 3 strains of the regulons involved in virulence shows that they present a similar pathogenic strategy to strains belonging to clades 1 and 2, even though they have been genetically isolated. In this respect, it is important to highlight that clade 3 strains are members of the *P. aeruginosa* species, even though their genetic divergence is high when compared to the sequence conservation of other phylogroups. However, even though they have a slightly lower ANI than that accepted for members of a bacterial species [41], the conservation of their chromosome synteny (www.pseudomonas.com; accessed on 15 May 2021), pathogenic mechanisms, and other aspects of their biology show that they indeed belong to this species.

### 3.3. Analysis of the Genomes of Strains Belonging to Clade 5

Reboud et al. reported in 2016 [35] that there were two groups of *exlBA*-containing *P. aeruginosa* strains, and that each group had a distinct type of deletion of genes encoding the T3SS. In 2019, Freschi et al. [17] reported the existence of clade 5, which was closely related to clades 1 and 2 but shared several genomic characteristics with clade 3–including the lack of genes involved in T3SS, and of the *rhlC* and *phzH* genes. It is possible that phylogroup 5 is derived from clade 2 since there is a vestige of the PA14_14300 gene (46 bp) that is present in the chromosomal region where PA14_14300, PA14_14310, and PA14_14320 are encoded in strain PA14 (strains presenting this PA14_14300 fragment are shown at the end of Table S2 and are marked in red).

To analyze the genomic context of these reported deletions, we used the leg-wound isolate PA-W1 strain that was previously reported to belong to clade 5 [17]. We found that the *exlBA* operon was present in the same genomic context as in strains belonging to clade 3, linked to the *phhABC* operon (www.pseudomonas.com; accessed on 15 May 2021), but the deletions affecting genes encoding proteins of the T3SS, *rhlC,* and *phzH* were completely different (Figure 3).

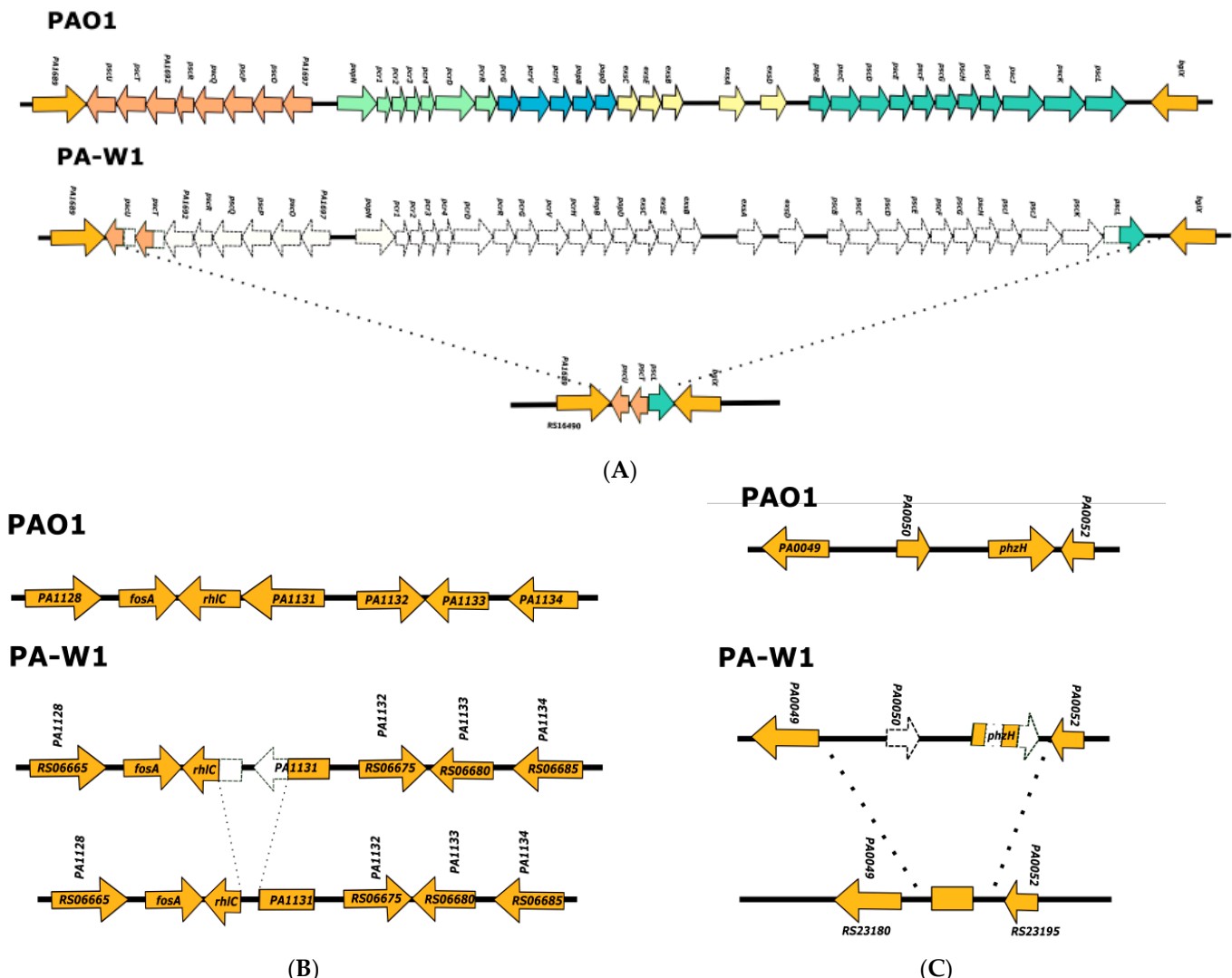

**Figure 3. Analysis of PA-W1 genome regions where the genes involved in T3SS, *rhlC*, and *phzH* are encoded in the PAO1 strain.** (**A**) Representation of the PAO1 and PA-W1 chromosomal regions where genes involved in the T3SS are encoded in the former strain; (**B**) representation of the PAO1 and PA-W1 chromosomal regions where *rhlC* is encoded in the former strain; (**C**) representation of the PAO1 and PA-W1 chromosomal regions where *phzH* is encoded in the former strain. Color codes used are the same as in Figure 2.

To search for strains belonging to clade 5 among the 4955 genomes deposited in PsGenDatab, we used either the 42 bp sequence that was created when 33 of the 36 T3SS genes were deleted (Figure 3), or the 30 bp sequence that was formed when part of the *rhlC* and the gene corresponding to the PAO1 PA1131 gene were deleted (Figure 3). In both cases, 20 *exlBA* positive strains were detected (see Supplementary Information Table S5). These 20 strains belonging to clade 5 share a high degree of nucleotide conservation with strains of clades 1 and 2 (with an ANI value of more than 97, Table 2), which is consistent with previous reports [16,17].

The most plausible explanation for the origin of strains belonging to clade 5 (Figure 4) is that a strain belonging to clade 2 inherited the *exlBA* operon from a strain of clade 3 by homologous genetic recombination with sequences flanking the exolysin A coding genes, since the *exlBA* operon is encoded in the same genomic context in both clades. However, when the *exlBA* operon was inherited, deletions of genes encoding the T3SS and of the *rhlC* and *phzH* genes were selected (Figure 3). These deletions were independent events to

those giving rise to clade 3, and their independent selection shows that a strong selective pressure for the loss of these genes was exerted for these clades to persist.

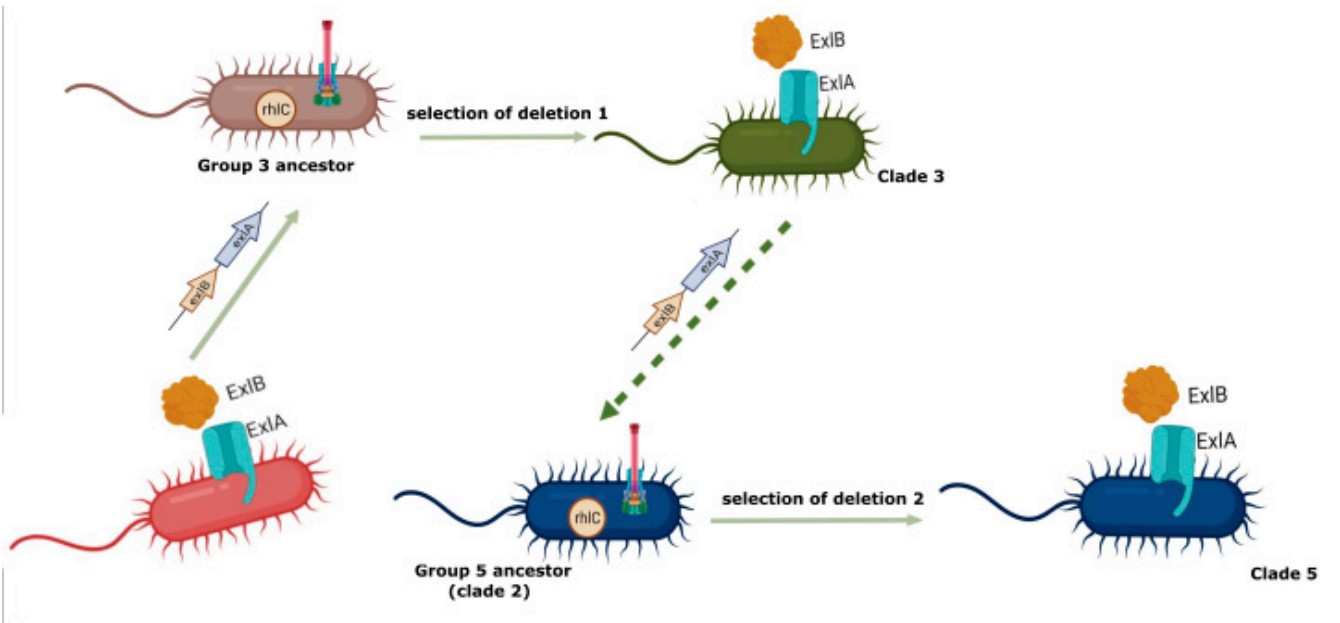

**Figure 4. Schematic representation of the proposed hypothesis for the origin of strains belonging to clades 3 and 5.** We propose that the *exlBA* operon was transferred by HGT from a non-related bacteria to the *P. aeruginosa* ancestor of clade 3, and that the inheritance of these genes encoding for the production of exolysin A was incompatible with the production of the T3SS, so the deletion of the five operons encoding this virulence factor was selected; concomitantly in order to compensate for the lack of T3SS the deletion of *rhlC* and of *phz* were selected. In turn, phylogroup 5 was originated by the HGT from a *P. aeruginosa* clade 3 strain of the *exlBA* operon to a *P. aeruginosa* strain possibly belonging to clade 2, and the deletion of genes encoding for the T3SS, *rhlC*, and *phzH* were selected in independent events. This image was created by biorender.com.

As mentioned, the production of exolysin A appears to be incompatible with the presence of the T3SS because no *P. aeruginosa* strains have been isolated that produce both systems that cause a cytotoxic effect (Figure 4). The fact that strains belonging to clade 5 present an independent deletion event of genes encoding T3SS reinforces this conclusion.

We propose that the deletion of *rhlC* and *phzH* in strains belonging to clades 3 and 5 are secondary events to the inheritance of *exlBA* and the concomitant loss of genes encoding the T3SS, since this secretion system is one of the most important *P. aeruginosa* virulence factors [11], and so the exolysin A production might not be able to fully compensate the attenuation of strains having a deletion of the T3SS genes. Furthermore, the existence of unrelated deletions in the same genes in phylogroups 3 and 5 strongly suggest that the rearrangement of the pathogenicity-related traits is the most plausible response for the expression of virulence in the context of exolysin A production.

Another important virulence factor, besides T3SS, is the oxidizing toxin PYO. The deletion of *phzH* is expected to cause increased production of this phenazine by redirecting the carbon flow to its synthesis [19]. In the case of the deletion of *rhlC*, it has been reported that PAO1 mutants lacking this gene overproduce PYO [18] by a not yet understood molecular mechanism. Thus, the deletion of *phzH* and *rhlC* is expected to cause an increment in PYO production that could compensate for the lack of a functional T3SS in strains belonging to clades 3 and 5.

It has recently been reported that *P. aeruginosa* strains that produce exolysin A synthesize higher levels of PYO and higher biofilm mass [42], an observation that is consistent with the proposed hypothesis.

### 3.4. Searching for Clade 4

The existence of clade 4 was proposed by Freschi et al. in 2019 when they analyzed 1311 high-quality *P. aeruginosa* genomes [17]. The phylogeny reported in their article gives solid evidence for the existence of five groups, and they reported matrices of the presence of several pathogenicity-related markers that were used in our analysis. Besides the previously analyzed phylogroups, they defined clade 4, which is closely related to clades 1, 2 and 5, and is characterized by the absence of *mexF*, *cat*, *oprA*, *triC*, *PDC-8*, and *pvrR*, and by the presence of *exoS*, *PDC-1*, *arnA*, *fosA*, *pmrC*, and *pmrF*. Ozer et al. in 2019 [16] did not detect this phylogroup among the genomes of the 739 strains they analyzed, presumably because they used a smaller number of genomes.

To search for a genomic marker of phylogroup 4 that would enable us to search for members of this clade among the 4955 genomes deposited in the PsGenDatab, we analyzed the genome of strain AUS485, which is a member of this group [17], but we could not detect any marker that could be used for the focused analysis that we performed in this work.

To identify members of clade 4 among the 4955 *P. aeruginosa* genomes deposited in PsGenDatab, we searched for genomes with the mentioned genetic markers reported for clade 4 [17]. To determine whether these strains grouped with strain AUS485 and thus could be considered as members of clade 4, we made a phylogenetic tree with known members of phylogroups 1 and 2 and used PA7 as an outgroup (Figure 5). We found that strain AUS485 did branch outside clades 1 and 2 as expected, but the strains that we detected as candidates for being members of clade 4 formed part of clade 2. Thus, using the approach of tracking some genetic markers that were previously identified by the analysis of whole genome sequences [16,17], we could not detect any member of clade 4 besides AUS485, and we are unable to make any hypothesis on the origin of phylogroup 4.

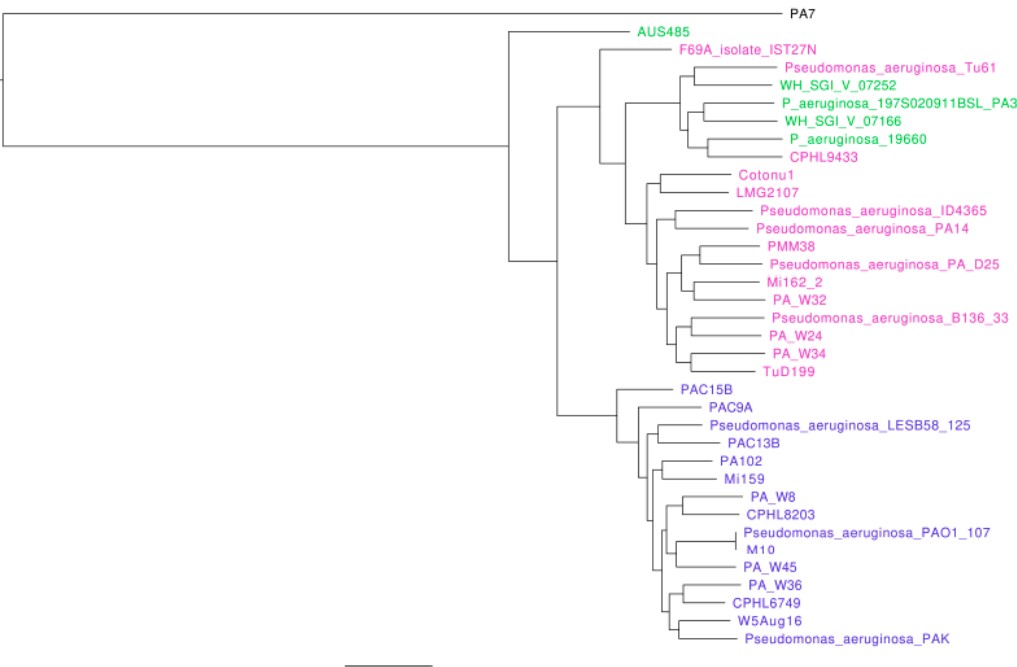

**Figure 5. Phylogenetic tree of the whole genome sequence of strains belonging to clade 1, clade 2, and of the putative members of clade 4**. To construct the phylogenetic tree, we retrieved from the NCBI the genome sequences of *P. aeruginosa* members from clades 1 (shown in blue) and 2 (shown in purple), and candidates of strains belonging to clade 4 (shown in green), as well as PA7 used as outgroup. The accession numbers of the genomes of all strains used are provided in the materials and methods section.

## 4. Concluding Remarks

The ability of *P. aeruginosa* to establish pathogenic interactions is a central part of its lifestyle [1,3]. Thus, genes involved in pathogenicity form part of its core genome [8,9,17], and three of the 10 genes that explain the genetic variation of strains belonging to clades 1 and 2 are involved in its pathogenicity [43].

The identification of four or five phylogroups within *P. aeruginosa* species by the analysis of hundreds of genomes [16,17] is an important achievement, which could considerably impact the treatment of infections caused by this bacterium, since research areas such as epidemiology, antibiotic resistance, molecular mechanisms involved in pathogenicity, and the evolution of this bacterial species are impacted [44]. However, efforts to find correlations between strains belonging to any particular clade as being preferentially clinical or environmental, or showing a particular geographical distribution, have been unsuccessful [16,17]. The only reported endemicity of *P. aeruginosa* strains has been a subclade of phylogroup 2 strains that were isolated in Cuatro Ciénegas, an oasis in the Chihuahuan desert in Mexico, during a drought crisis [45]. In addition, it was reported that there is a statistically significant predominance of strains isolated from cystic fibrosis (CF) patients among strains belonging to clade 1, and that strains belonging to clade 2 were more likely to produce eye infections [16].

Clades 3 and 5 are characterized by the production of ExlA and their lack of T3SS, and they represent a very small fraction of strains that have been analyzed in previous reports [16,17], and of the 4955 genomes that have been deposited in the PsGenDatab, according to the results presented here (36 and 20 strains, respectively). These results could be interpreted as if these clades lacked importance, but the low frequency of the isolates belonging to phylogroups 3 and 5 might arise as a bias in the sampling of sequenced genomes, as is suggested by the high frequency of exolysin A producing *P. aeruginosa* strains (six out nine) among isolates from healthy farm animals [42]. It is thus important to screen different clinical and environmental settings to have a more complete picture of the incidence of strains belonging to these phylogroups.

Our approach of screening for genomic markers that are characteristic of each *P. aeruginosa* clade, which could only be made using the solid phylogenies of this bacterium species that were previously reported [16,17], was successful in distinguishing members of clades 1, 2, 3, and 5 among the 4955 genomes deposited in the PsGenDatab and enabled us to propose hypotheses on the origins of these phylogroups. We propose that clade 2 diverged from clade 1 by the deletion of *exoS*, the acquisition of three genes in the same chromosomal region (PA14_14320, PA14_14310, and PA14_14300), and the inheritance of the PAPI-2 genomic island which encodes ExoU and SpcU, among other genes (Figure 1). The fixation of the clone giving rise to clade 2 seems to be due to the low recombination frequency between these two clades, caused by an unknown mechanism [16]. However, the genome conservation between these two phylogroups is remarkably high, measured by the ANI index of some strains (Table 2), or using 541 strains of group 1 and 186 strains of group 2 [16]. In both instances, ANIs of more than 98% were found between these clades, and between members of clade 2, only slightly lower than the more than 99% ANI presented among the strains belonging to clade 1 (Table 2).

In the case of clade 3, we identified 36 strains belonging to this phylogroup in the PsGenDatab using either the 53 bp PA7 sequence that is present instead of the 36 PAO1 genes encoding the T3SS, or the 93 bp sequence that is in the place of the *rhlC* and surrounding genes in the PAO1 genome. All the 36 strains identified encode the *exlBA* operon linked to the *phhABC* operon that in PAO1 corresponds to the PA0872, PA0871, and PA0872 genes, and also contain a deletion of the *phzH* gene (Figure 2). It is likely the ancestor of clade 3 strains inherited the *exlBA* operon by HGT, and that the presence of this exolysin is incompatible with the T3SS, so deletion of the genes encoding this secretion system was selected. We propose that to increase the production of PYO, the deletion of *rhlC* and *phzH* were also selected (Figure 4), since PAO1 mutants in these genes hyperproduce PYO [18,19].

The case of strains belonging to clade 5 represents an independent event of formation of a *P. aeruginosa* phylogroup that has a similar pathogenic strategy as clade 3 strains that is based on the production of exolysin A, and that lacks RhlC and PhzH. The ANI value of more than 97% presented by phylogroup 5 compared to clades 1 and 2 (Table 2) is clearly within the *P. aeruginosa* species, but it is expected that the molecular mechanisms involved in their pathogenic interactions are common with the distantly related clade 3. There have been no reports studying this phenomenon, which might represent a case of convergent molecular evolution.

The analysis presented in this work opens an interesting field of research on the epidemiology and evolution of the bacterial opportunistic pathogen *P. aeruginosa* that identified some genetic markers characteristic of four previously described phylogroups that were identified by the analysis of whole-genome sequences. Several questions arise from the focused analysis that we have presented, such as which is the molecular mechanism involved in the reduced recombination between members of clades 1 and 2, understanding the particularities of the clades 3 and 5 pathogenicity strategies, or whether there is a high prevalence of these exolysin A producing phylogroups associated with a particular niche or type of infection, among others. These fundamental questions can be experimentally validated to contribute to the understanding of the biology of this fascinating bacterium and to the medical problems that it causes.

**Supplementary Materials:** The following supporting information can be downloaded at: https://www.mdpi.com/article/10.3390/d14050345/s1: Table S1: List of strains that contain *exoS* in the PsGenDatab; Table S2: List of strains that lack *exoS* and contain PA14_14300, that are deposited in the PsGenDatab; Table S3. ANI value of the 8 strains that contain *exoS* and *exoU*, and PA14_14300 compared with clade 1 and clade 2 strains. Table S4: List of strains that contain *exlBA* and contain deletions of T3SS encoding genes, *rhlC* and *phzH* as described in Figure 2, that are deposited in the PsGenDatab; Table S5: List of strains that contain *exlBA* and deletions of T3SS encoding genes, *rhlC* and *phzH* as described Figure 3, that are deposited in the PsGenDatab.

**Author Contributions:** Conceptualization, G.S.-C., S.E.Q.-M., L.S.-G., S.G.-R. and G.Y.P.-S.; methodology, S.E.Q.-M., S.G.-R. and G.Y.P.-S.; software, G.Y.P.-S.; investigation G.S.-C., S.E.Q.-M., S.G.-R. and G.Y.P.-S.; resources, G.S.-C.; data curation, S.E.Q.-M.; writing—original draft preparation, G.S.-C.; writing—review and editing, L.S.-G., S.E.Q.-M., S.G.-R. and G.Y.P.-S.; funding acquisition, G.S.-C. All authors have read and agreed to the published version of the manuscript.

**Funding:** G.S.-C. laboratory is supported in part by grant IN201222 from Programa de Apoyo a Proyectos de Investigación e Innovación Tecnológica, (Dirección General de Asuntos del Personal Académico–UNAM).

**Institutional Review Board Statement:** Not applicable.

**Data Availability Statement:** Not applicable.

**Acknowledgments:** S.E.Q.-M. is a doctorate student from Programa de Posgrado en Ciencias Bioquímicas, UNAM, and she received a CONACYT fellowship (CVU 1044595).

**Conflicts of Interest:** The authors declare no conflict of interest.

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
