# Peer review of "Tracking the Origins of Pseudomonas aeruginosa Phylogroups by Diversity and Evolutionary Analysis of Important Pathogenic Marker Genes†"

_diversity, doi:10.3390/d14050345_

Round 1

Reviewer 1 Report

Please include 'hypothetical' or another similar word in the title to indicate the content of this manuscript.

Line 168, spelling error, search

Lines 169 - 171, throughout the text, there are places where I would have assumed the sentence would be written in the past tense.

Line 207 - 211, this part is confusing, please rewrite to clarify if 'A strain' is from clade 2?

Line 296 - 302, This is a very long sentence. Please split the sentence and rewrite for clarity.

Line 357, typo = et al.

Line 361, typo = et al.

Line 397, typo = bp

Line 401, typo = bp

Line 499 - 501, For clarity, please consider listing these 'questions'

Author Response

Dear Reviewer 1, 

Thank you very much for the revision of our manuscript, we hace corrected all the mistakes you point out and addressed your concerns.

The paragraph where you asked us to define the questions, now reads:

Several questions arise from the analysis that we have presented, such as which is the molecular mechanism involved in the reduced recombination between members of clades 1 and 2, understanding the particularities of the clades 3 and 5 pathogenicity strategies, or whether there is a high prevalence of these exolysin A producing phylogroups associated with a particular niche or type of infection, among others. These fundamental questions can be experimentally validated to contribute to the understanding of the biology of this fascinating bacterium and to the medical problems that it causes. 

Reviewer 2 Report

The manuscript presents the evaluation of the diversity and evolution of the phylogroups of pathogenic P. aeruginosa based on the occurrence and diversity of central pathogenic marker genes. The study however focused almost completly on pathogenic or opportunitsic pathogenic strains of P. aeruginosa and environmental and plant-growth promoting / biocontrol or plant pathogenic strains are not included. This focus on pathogenic P. aeruginosa strains should have been clearly emphasized - even in the title, like: "Tracking the origins of pathogenic P. aeruginosa phylogroups by diversity and evolutionary analysis of important pathogenic marker genes." Intentionally, the study did not apply whole-geneome based sequence analysis of the selected strains and is therefore limited to a narrow view of the diversity of pathogenic strains. In contrast, in the well-known case of Burkholderia, which included pathogenic, opportunistic pathogenic, environmental and plant-associated/beneficial and even symbiotic strains, whole-genome sequences and conserved sequence indels (CSI) were successfully used as alternative approach to create molecular markers for the demarcation of different phylogroups/ new species of Burkholderia (see Sawana et al., 2014; Dobritsa et al., 2016). In my view, the more wide approach of novel diversity and evolution analysis should have been used. However, if the narrow, focussed approach is accepted, the manuscript has its merits. However, a modified title should be used and the intention to exclude whole-genome analysis should be clearly expressed.

Author Response

Dear Reviewer 2,

Thank you very much for reviewing our work and making valuable suggestions.

According to your suggestion we changed the title of our work to:

Tracking the origins of Pseudomonas aeruginosa phylogroups by diversity and evolutionary analysis of important pathogenic marker genes  

We did not include restricted our analysis to pathogenic strains of P. aeruginosa, even though the majority of the 4955 sequenced-genomes deposited in the Pseudomonas Genome Database are from clinical origin. Furthermore, comparative genomic analysis of environmental and clinical isolates, have shown that they do not constitute distinct populations. There are not defined P. aeruginosa pathovars, and all strains are potentially able to infect different hosts. In addition, plant promoting strains do not represent a particular phylogroup. We modified different parts of the text to clarify these peculiar genomic characteristics of P. aeruginosa and to address your concern. The main parts of the manuscripts that were modified are:

Lines 35-39.

The genomic constitution of this bacterial species is characterized by the high degree of genome conservation among clinical and environmental isolates that have different geographical origins and times of isolation, and the conservation among all isolates, both environmental and clinical, of the virulence-associated traits [8, 9]; thus, all strains are potentially pathogenic.

Lines 71-80.

The aim of this work is the detailed bioinformatic analysis of the chromosomal regions encoding the distinct characteristics of each of the four phylogenetic groups reported by Olzer et al in 2019 [16] using representative strains to define marker sequences, that were then used to classify the 4955 P. aeruginosa genomes deposited in the Pseudomonas Genome Database (PsGenDatab, www.pseudomonas.com), which includes both clinical as environmental isolates [20]. The results obtained by this analysis enabled us to propose hypotheses on the origins of four P. aeruginosa phylogroups. 

Lines 503-505.  

The analysis presented in this work opens an interesting field of research on the epidemiology and evolution of the bacterial opportunistic pathogen P. aeruginosa that is not dependent on the analysis of whole-genome sequences.

Reviewer 3 Report

The manuscript titled “Tracking the origins of Pseudomonas aeruginosa phylogroups:  A bioinformatic approach” is devoted to analysis of its phylogeny based on the whole genome sequences of hundreds of isolates. These clades contain both clinical and environmental strains,  which show no particular geographical distribution. The major phylogroups (clades 1 and 2) were characterized by the nearly mutually exclusive production of the virulence effectors secreted by the type three secretion system (T3SS) ExoS and ExoU, respectively. Clade 3 is the most genetically diverse and shares with clade 5, which is closely related to clades 1 and 2, the production of the pore-forming exolysin A and the lack of T3SS, among other characteristics.  4955 P. aeruginosa genomes deposited in the Pseudomonas Genome database were analyzed and the origins of four of the five phylogroups of this bacterial species were postulated based on new hypothesis.

The manuscript is generally well written, except a few mistyping, like at line 83: “The search of exoS among the geomes deposited in the PsGenDatab …”

But, there are several significant problems:

  • in #2.1. Analysis of genome sequences (line 77)

“The PsGenDatab (www.pseudomonas.com) was used to obtain genome sequences used in this work and to carry out blast searches. P. aeruginosa strains PAO1, PA14, PA7 80 y PA-W1 were used as references for clades 1, 2, 3 and 5, respectively. Global alignments were carried out by EMBL-EBI sequence analysis tools Clustal Omega [21]. “

There is no sufficient description of analysis method  in Materials and Methods #2.1.

It is clear from the following text of Results and Discussion that to get information about the chromosomal differences of strains belonging to clades 1 and 2, the chromosomal regions that surround the exoS (PA3841) and exoU (PA14_51530) genes, the Authors analysed by BLAST  all other genomes.  Thus, the analysis was limited by a short genomic region surrounding previously described marker genes.

As it was stated at line 439:

“Thus, genes involved in pathogenicity form part of its core genome ...  and 3 of the 10 genes that explain the genetic variation of strains belonging to clades 1 and 2 are involved in its pathogenicity...”

As it was described in the cited work of Freschi L, et al. (2019),  665-gene P. aeruginosa core genome was found from the data set of 1,311 high-quality genomes of Pseudomonas aeruginosa, which constitutes only 1% of the entire pan-genome, and  it represents a conservative estimate of the actual core genome. The phylogeny based on this core genome provides strong evidence for a five-group population structure that includes two previously undescribed groups of isolates.

Such highly reliable results are criticized by the Authors of current manuscript at Line 357:

“The existence of clade 4 was proposed by Freschi et al in 2019 when they analyzed 1311 high quality P. aeruginosa genomes. … According to these authors, clade 4 is closely related to clades 1, 2 and 5, and is characterized by the absence of mexF, cat, oprA, triC, 359 PDC-8, and pvrR, and the presence of exoS, PDC-1, arnA, fosA, pmrC and pmrF. Strain AUS485 was reported to belong to this clade. Ozer et al, 2019, did not detect any sign of this phylogroup among the genomes of the 739 strains they analyzed.”

“To search for a genomic marker of phylogroup 4 we analyzed the genome of strain  AUS485 starting with the exoS region and found that this strain lacks this gene and presents a homolog of PA14_14300, suggesting that it is derived from a clade 2 ancestor. We  concluded that it was mistakenly reported as having an exoS gene because it contains exoT  which is highly homologous.”

“To identify members of clade 4 among the 4955 P. aeruginosa genomes deposited in PsGenDatab we searched for genomes with the mentioned genetic markers reported for the clade 4, except for exoS and found 4 candidates.”

“To determine whether these  strains grouped with strain AUS485 and thus could be considered as members of clade 4,  we made a phylogenetic tree with known members of phylogroups 1 and 2 and used PA7 372 as an outgroup (Fig. 5). “

“We found that strain AUS485 did branch outside clades 1 and 2, but the strains that we detected as candidates for being members of clade 4 formed part of clade 2. Thus, we cannot make any hypothesis on the origin of phylogroup 4, since the approach we followed for the analysis of P. aeruginosa phylogroups did not enable us to detect strains belonging to this clade in the PsGenData”.

Reviewer cannot agree to the logic of discussion, because Freschi et al (2019) applied two methods - SNP analysis  (n= 55,664) of  448-gene core genome and  the genome architecture calculated using flexible gene presence/absence, n=26,420 genes. The Authors used a few marker genes and ANI comparison for P. aeruginosa strains to develop the following hypothesis:

Line 468. “We propose that clade 2 diverged from clade 1 by the deletion of exoS, the acquisition of three genes in the same chromosomal region (PA14_14320, 469 PA14_14310, and PA14_14300) and the inheritance of the PAPI-2 genomic island which encodes ExoU and SpcU, among other genes (Fig. 1). The fixation of the clone giving rise to clade 2 seems to be due to the low recombination frequency between these two clades caused by an unknown mechanism ... However, the genome conservation between these two phylogroups is remarkably high measured by the ANI index of some strains (Table 2) or using 541 strains of group 1 and 186 strains of group 2..”

It is obvious that phylogeny of a few marker genes is not a phylogeny of Pseudomonas aeruginosa.

Such important hypothesis must be justified by several different methods, including all methods describe in updated phylogenetic analysis of Pseudomonas aeruginosa (SNP of core genome, using flexible gene presence/absence, MLST, and ANI).  

The manuscript needs significant additional analysis of data

Freschi L, Vincent AT, Jeukens J, Emond-Rheault JG, Kukavica-Ibrulj I, Dupont MJ, Charette SJ, Boyle B, Levesque RC. The Pseudomonas aeruginosa pan-genome provides new insights on its population structure, horizontal gene transfer, and pathogenicity. Genome biology and evolution. 2019 Jan;11(1):109-20.

Author Response

Dear Reviewer 3,

First, we want to thank you for your comments and suggestions. We consider that our article has been very much improved by including the modifications that clarify your queries.

You accurately detected that our manuscript was written in a way that could be interpreted as a criticism for the work of Freschi at al. published in 2019, but that was not our intention at all. On the contrary, the analysis that we report in this manuscript uses as one of its main inputs the seminal work of this article. To avoid this confusion we modified our manuscript, so it is clear that we are only making a focused analysis of some genetic markers that were detected in the Freschi et al., 2019 article, and that our approach does not aim to question the phylogeny described by these authors. This approach was not only helpful in allowing us to formulate plausible hypotheses for the origins of four of the five P. aeruginosa phylogroups, but also provided a simple way to assign genomes to these phylogroups without the need to carry out more complex and computationally demanding bioinformatic analyses.

In the new version of our manuscript besides pointing out that we did not analyze the phylogeny of P. aeruginosa, we clearly state that our analysis relies on the work published by Freschi et al, (2019) and by Ozer et al. (2019). We emphasize that we are not questioning the existence of clade 4 (the methods used did not enabled us to do it and it was not the aim of our analysis), but that our approach did not allow us to find a simple genetic marker to associate and detect strains belonging to this phylogroup in the Pseudomonas Genomes Database (PsGenDatab).

We agree with you that if we pretend to do a phylogenetic analysis using the 4955 whole genome sequences deposited in PsGenDatab, we should use more robust bioinformatic methods devised for this type of analysis, as those used by Freschi et al, (2019) and by Ozer et al. (2019). However, this has never been our goal, and that is why we only performed the basic analysis described in section 2.1 (which has been modified according to your concerns).

The title of the new version of our work was modified, according to another reviewer suggestion, but the new title “Tracking the origins of Pseudomonas aeruginosa phylogroups by diversity and evolutionary analysis of important pathogenic marker genes”, more clearly define that we did not intend to review the phylogeny of this bacterium, but to track some specific markers among the previously defined phylogroups.

The paragraphs that were modified in the new version to address your concerns are:

Last paragraph of the Introduction section:

“The aim of this work is the bioinformatic analysis of the chromosomal regions encoding the distinct characteristics of each of the phylogenetic groups reported by Olzer et al in 2019 [16] and Freschi et al in 2019 [17], using representative strains to define marker sequences, that were then used to classify the 4955 P. aeruginosa genomes deposited in the Pseudomonas Genome Database (PsGenDatab, www.pseudomonas.com), which includes both clinical as environmental isolates [20]. We used as a starting point for our analysis two valuable resources: On the one hand, the huge amount of information organized and curated in PsGenDatab and on the other the whole genome analysis of hundreds of strains already reported [16, 17] that defined the P. aeruginosa pangenome [17] and which allowed us to track specific genetic markers related to pathogenicity that were defined in these works. The results obtained by this focused analysis enabled us to propose hypotheses on the origins of four of the five P. aeruginosaphylogroups. In addition, we tried to follow a similar strategy to detect strains belonging to the group defined by Freschi et al., 2019 [17] as group 4, which was identified by whole sequence analysis, but the focused strategy presented here did not result in the identification of genetic markers that could be used in tracking clade 4 among the genomes deposited in PsGenDatab.”

 Section 3.1.4 Searching for clade 4.

“The existence of clade 4 was proposed by Freschi et al. in 2019 when they analyzed 1311 high-quality P. aeruginosa genomes [17]. The phylogeny reported in their article gives solid evidence for the existence of five groups, and they reported matrices of the presence of several pathogenicity-related markers that were used in our analysis. Besides the previously analyzed phylogroups, they defined clade 4 which is closely related to clades 1, 2 and 5, and is characterized by the absence of mexF, cat, oprA, triC,PDC-8, and pvrR, and by the presence of exoS, PDC-1, arnA, fosA, pmrC and pmrF. Ozer et al. in 2019 [16] did not detect this phylogroup among the genomes of the 739 strains they analyzed, presumably because they used a smaller number of genomes.

To search for a genomic marker of phylogroup 4 that would enable us to search for members of this clade among the 4955 genomes deposited in the PsGenDatab, we analyzed the genome of strain AUS485 which is a member of this group [17], but we could not detect any marker that could be used for the focused analysis that we performed in this work.

To identify members of clade 4 among the 4955 P. aeruginosa genomes deposited in PsGenDatab we searched for genomes with the mentioned genetic markers reported for clade 4 [17]. To determine whether these strains grouped with strain AUS485 and thus could be considered as members of clade 4, we made a phylogenetic tree with known members of phylogroups 1 and 2 and used PA7 as an outgroup (Fig. 5). We found that strain AUS485 did branch outside clades 1 and 2 as expected, but the strains that we detected as candidates for being members of clade 4 formed part of clade 2. Thus, using the approach of tracking some genetic markers that were previously identified by the analysis of whole genome sequences [16, 17], we could not detect any member of clade 4 besides AUS485, and we are unable to make any hypothesis on the origin of phylogroup 4.”

Section 4. Concluding remarks. (fourth and final paragraphs)

“Our approach of screening for genomic markers that are characteristic of each P. aeruginosa clade, which could only be made using the solid phylogenies of this bacterium species that were previously reported [16, 17] was successful in distinguishing members of clades 1, 2, 3 and 5 among the 4955 genomes deposited in the PsGenDatab and enabled us to propose hypotheses on the origins of these phylogroups…”

“The analysis presented in this work opens an interesting field of research on the epidemiology and evolution of the bacterial opportunistic pathogen P. aeruginosa that identified some genetic markers characteristic of four previously described phylogroups that were identified by the analysis of whole-genome sequences. Several questions arise from the focus analysis that we have presented, such as which is the molecular mechanism involved in the reduced recombination between members of clades 1 and 2, understanding the particularities of the clades 3 and 5 pathogenicity strategies, or whether there is a high prevalence of these exolysin A producing phylogroups associated with a particular niche or type of infection, among others. These fundamental questions can be experimentally validated to contribute to the understanding of the biology of this fascinating bacterium and to the medical problems that it causes.”

Round 2

Reviewer 3 Report

The manuscript "Tracking the origins of Pseudomonas aeruginosa phylogroups by diversity and evolutionary analysis of important pathogenic marker genes" is reporting very important information. All questions of reviewer were answered, and manuscript can be published.